# Measuring Glycolytic Activity with Hyperpolarized [^2^H_7_, U-^13^C_6_] D-Glucose in the Naive Mouse Brain under Different Anesthetic Conditions

**DOI:** 10.3390/metabo11070413

**Published:** 2021-06-23

**Authors:** Emmanuelle Flatt, Bernard Lanz, Yves Pilloud, Andrea Capozzi, Mathilde Hauge Lerche, Rolf Gruetter, Mor Mishkovsky

**Affiliations:** 1Laboratory of Functional and Metabolic Imaging, École Polytechnique Fédérale de Lausanne (EPFL), 1015 Lausanne, Switzerland; emmanuelle.flatt@epfl.ch (E.F.); bernard.lanz@epfl.ch (B.L.); andrea.capozzi@epfl.ch (A.C.); rolf.gruetter@epfl.ch (R.G.); 2Center for Biomedical Imaging (CIBM), École Polytechnique Fédérale de Lausanne (EPFL), 1015 Lausanne, Switzerland; yves.pilloud@epfl.ch; 3Department of Health Technology, Technical University of Denmark, Oersteds Pl. Bldg. 349, Room 114, 2800 Kgs. Lyngby, Denmark; mhauler@dtu.dk

**Keywords:** cerebral metabolism, MRI, DNP, DMRS, ^1^H MRS, ^13^C MRS, CMR_Glc_, metabolic model

## Abstract

Glucose is the primary fuel for the brain; its metabolism is linked with cerebral function. Different magnetic resonance spectroscopy (MRS) techniques are available to assess glucose metabolism, providing complementary information. Our first aim was to investigate the difference between hyperpolarized ^13^C-glucose MRS and non-hyperpolarized ^2^H-glucose MRS to interrogate cerebral glycolysis. Isoflurane anesthesia is commonly employed in preclinical MRS, but it affects cerebral hemodynamics and functional connectivity. A combination of low doses of isoflurane and medetomidine is routinely used in rodent functional magnetic resonance imaging (fMRI) and shows similar functional connectivity, as in awake animals. As glucose metabolism is tightly linked to neuronal activity, our second aim was to assess the impact of these two anesthetic conditions on the cerebral metabolism of glucose. Brain metabolism of hyperpolarized ^13^C-glucose and non-hyperpolaized ^2^H-glucose was monitored in two groups of mice in a 9.4 T MRI system. We found that the very different duration and temporal resolution of the two techniques enable highlighting the different aspects in glucose metabolism. We demonstrate (by numerical simulations) that hyperpolarized ^13^C-glucose reports on de novo lactate synthesis and is sensitive to cerebral metabolic rate of glucose (CMR_Glc_). We show that variations in cerebral glucose metabolism, under different anesthesia, are reflected differently in hyperpolarized and non-hyperpolarized X-nuclei glucose MRS.

## 1. Introduction

The mammalian brain is mainly supported by glucose (Glc) as fuel to meet its high metabolic demand [1], consuming up to 25% of circulating Glc under normal conditions [2]. In physiological conditions, Glc metabolism is tightly linked with cerebral function [3,4]. Molecular imaging enables visualization and quantification of the function of cerebral biological processes, bringing essential new perspectives for disease assessment [5]. Because aberrant Glc metabolism is a key factor in many neurological disorders [6,7,8], Glc is the substrate of choice for metabolic imaging. Various techniques are available to study in vivo glycolytic metabolism non-invasively, giving complementary information on the metabolic pathways of Glc. Positron emission tomography (PET) of the radioactive ^18^F labeled fluorodeoxyglucose (FDG) [9,10], widely employed clinically, is capable to monitor uptake of this Glc analog with high sensitivity and good spatial resolution. However, it is not possible to distinguish between the tracer and its phosphorylated form, and this tracer cannot inform on the metabolic pathways subsequent to Glc phosphorylation. In addition, the high physiologic brain glucose consumption leads to strong background uptake of ^18^F-FDG, thus limiting its application for neuroimaging [11]. Additionally, a substantial radiation exposition dose during PET scans may prohibit the possibility for multiple repeated measurements of the same individual to avoid overexposure to radiation. Magnetic resonance (MR) provides various methodological approaches to interrogate Glc metabolism non-invasively and with non-ionizing radiation. Glc uptake can be monitored by indirect detection of the glucose’s labile protons using chemical exchange saturation transfer (CEST) MRI upon infusion of non-labeled Glc [12,13,14]. Nonetheless, spectral overlaps may lead to the contribution of other metabolites to change its image contrast, and the dependency of the exchange rate on pH [15] may bias the results. Magnetic resonance spectroscopy (MRS) measurements of non-hyperpolarized (non-HP; i.e., thermally polarized) ^13^C labeled Glc bring important kinetic information on brain metabolism with low background signal, and it enabled to demonstrate the compartmentation of the cerebral Glc metabolism between the neurons and the glia [16]. However, its low sensitivity necessitates long acquisition times and long infusion times, which makes it difficult to apply for human studies and not suitable for clinical applications, despite being an interesting preclinical research tool providing extensive biochemical information on brain metabolism in vivo. Given the shorter longitudinal relaxation times of ^2^H as compared to that of ^1^H or ^13^C spins, a faster signal averaging scheme can be applied with ^2^H MRS compared to ^13^C MRS thus increasing the sensitivity per unit of time [17]. Implementation of MRSI following infusion of non-HP [6,6′-^2^H_2_] Glc has shown its large potential by enabling imaging of Glc metabolism in the human brain [18] and may open new avenues for clinical diagnostic.

Dissolution dynamic nuclear polarization (dDNP) of metabolites dramatically enhances MR sensitivity, allowing to monitor real-time intermediary metabolism [19,20]. This dramatic increase in sensitivity paves the way to a new class of metabolic neuroimaging [21,22,23,24]. While most studies have mainly focused on the potential of HP [1-^13^C]pyruvate (Pyr), due to its excellent properties as dDNP molecular imaging contrast agent, dDNP is a versatile technique that can be employed to polarize a large variety of metabolically relevant molecules [25]. It was demonstrated that HP [^2^H_7_, U-^13^C_6_] Glc can be measured in vivo [26] and employed for the real time detection of glycolytic reactions in a lymphoma mouse model [27], in healthy mouse brain [28] and brain tumors [29]. The latter study has reported a discrepancy between the lactate (Lac) pool size measured by proton MRS and the labeling of Lac from HP [^2^H_7_, U-^13^C_6_] Glc, thus emphasizing the unique contrast with HP [^2^H_7_, U-^13^C_6_] Glc MRS, which enables monitoring the net ^13^C Lac production. Moreover, it was hypothesized that due to the very different acquisition time between X-nuclei MRS of HP ^13^C Glc and non-HP ^13^C Glc (seconds vs. hours, respectively), one should anticipate a difference in the labeling pattern between the two approaches [29]. Hence, the first aim of the present study was to further investigate the nature of these differences.

Brain consciousness influences cerebral metabolism, in particular, cerebral Glc metabolism, is closely linked to neuronal activity [4,30,31]. Anesthetics are commonly employed in pre-clinical in vivo MR studies and have been shown to impair neural transmission and to affect various aspects in rodent physiology in a dose-dependent manner [32,33,34,35,36]. Interestingly, while isoflurane is one of the most common anesthetic drugs used in preclinical MR metabolic studies, it is known to affect cerebral hemodynamics [32,33] and to influence functional connectivity [34]. Recent studies reported that combining isoflurane and medetomidine at levels of about half their mono-anesthetic dosages enables to maintain similar functional connectivity as in the awake rats, considerably better than either of the mono-anesthetic protocol alone, such as isoflurane [34,37]. The combination of medetomidine and isoflurane is now currently considered one of the most appropriate anesthetic protocols available for the evaluation of murine functional connectivity [36]. However, the extent to which different anesthesia protocols impact cerebral metabolism is an important topic that remains to be investigated [38]. Thus, the second aim was to assess the impact of this two anesthetic conditions on the cerebral metabolism of HP [^2^H_7_, U-^13^C_6_] Glc and non-HP [6,6′-^2^H_2_] Glc detected by X-nuclei MRS.

## 2. Results

### 2.1. Neurochemical Profile

Steady-state metabolite concentrations derived from the ^1^H spectra enable to evaluate the effect of the two anesthetic conditions on the endogenous concentration of brain metabolites. A significantly higher lactate pool-size (69%, *p* = 0.001, Figure 1a,d,g) and significantly lower concentration of phosphocreatine (−21%, *p* = 0.047) was found in animals anesthetized by isoflurane (ISO group) compared to those anesthetized by the combination of medetomidine and isoflurane (MED–ISO group). Creatine concentration was higher in the ISO group, without reaching significance, consequently, the calculated creatine-to-phosphocreatine ratio (Cr/PCr) was found to be significantly higher in the ISO group (60%, *p* = 0.041). All other metabolites in the neurochemical profile did not show any significant difference (Appendix A).

### 2.2. Measurement of Glc Metabolism by [6,6′-^2^H_2_] Glc MRS

A bolus of [6,6′-^2^H_2_] Glc in mice anesthetized by isoflurane resulted in the transfer of the ^2^H labeling to the glutamate–glutamine pool, water pool, and lactate pool as previously reported in the rat brain [17,18,39]. Interestingly, when a similar bolus was infused into mice anesthetized by a combination of medetomidine and isoflurane, no labeling in the lactate pool could be detected in the mouse brain (Figure 1b,e and Figure 2a,b).

The time courses of all animals in both groups were well reproducible (Figure 2e,f), and implies on a different systemic effect of the anesthesia on glucose metabolism.

Summed analyses (over the 64 min post glucose bolus) of the metabolite ratio showed a significantly higher HDO labeling in the ISO group compared to the MED-ISO group (+28.7 %, p = 0.043). Moreover, ^2^H Lac signal was detected only in animals anesthetized by isoflurane alone. A trend of higher glutamate–glutamine (Glx) labeling was observed in the MED–ISO group without reaching significance, while the kinetics of infused Glc was comparable in the two anesthetic conditions and did not reach any significant difference (Figure 2e,f and Appendix A). Blood glycemia and ^2^H glucose fractional enrichment that may influence these ratios were kept similar, and showed no significant differences between the groups (Appendix A).

### 2.3. Real-Time Measurement of HP [^2^H_7_, U-^13^C_6_] Glc Metabolism

Typical dynamic spectra following a bolus infusion of HP [^2^H_7_, U-^13^C_6_] Glc presented in Figure 2c,d demonstrate that under both anesthetic conditions [1-^13^C] Lac turnover can be readily detected. The Glc bolus and the produced Lac were detectable for 70 s post-injection. The [1-^13^C] Lac signal was detected for a longer period of time in the MED–ISO group (Figure 2h). The time courses of all animals in both groups were well reproducible (Figure 2g,h).

Total Lac-to-Glc ratio (LGR) was found significantly higher in the MED–ISO group compared to the ISO group (+128%, Figure 1c,f,i). Note that blood glycemia and ^13^C glucose fractional enrichment stayed similar between the two groups and showed no significant differences (Appendix A). Overall, these results indicate that medetomidine combined with a lower dose of isoflurane greatly increased the conversion of HP [^2^H_7_, U-^13^C_6_] Glc into its downstream metabolite Lac.

### 2.4. Numerical Simulation

The metabolic model for HP [^2^H_7_, U-^13^C_6_] Glc metabolism was simulated with a typical Glc input function reaching 70% fractional enrichment in 14 s and with a nominal brain Lac concentration of 4 μmol/g and cerebral metabolic rate of glucose (CMR_Glc_) of 0.5 μmol/g/min. The impact of the metabolic rate of Glc on the Lac ^13^C concentration was assessed by varying the CMR_Glc_ by steps of −50%, −20%, 0%, +20% and +100%, i.e., CMR_Glc_ = 0.25, 0.4, 0.5, 0.6, and 1 µmol/g/min (Figure 3c), while keeping the nominal Lac concentration of 4 μmol/g. The impact of the Lac pool size on the Lac ^13^C concentration was assessed by varying the total Lac concentration by steps of −50%, −20%, 0%, +20%, and +100%, i.e., [Lac] = 2.0, 3.2, 4.0, 4.8, and 8.0 µmol/g (Figure 3d), while keeping the nominal CMR_Glc_ of 0.5 μmol/g/min.

In a second step, the process was repeated by adding the effects of ^13^C HP signal losses due to T_1_ decay and repeated RF pulsing. Figure 3e shows the impact of these effects on the HP Glc signal, considering the same input function as in Figure 3b in terms of Glc ^13^C concentration. Figure 3f,g correspond to Figure 3c,d, respectively, with additional T_1_ decay and RF signal losses. Figure 3c,d illustrate the sensitivity of the HP ^13^C-Lac curves to the metabolic rate of Glc and the Lac pool size, respectively, in the current in vivo experimental protocol and for typical brain metabolic rates for Glc and Lac metabolism.

## 3. Discussion

In this study, we compared two recent MR-based methodologies to interrogate cerebral Glc metabolism, namely HP [^2^H_7_, U-^13^C_6_] Glc MRS and [6,6′-^2^H_2_] Glc MRS. While both techniques are based on detecting the chemical shift difference as the precursor (Glc) is metabolized into its downstream metabolites, the very different duration and temporal resolution of the two detection schemes highlight different aspects in Glc metabolism.

### 3.1. HP [^2^H_7_, U-^13^C_6_] Glc Is Sensitive to Different Metabolic Parameters Than Its Non-HP X-Nuclei MRS Counterparts

With the rapid HP signal decay, the ^13^C Lac turnover curve following HP [^2^H_7_, U-^13^C6] Glc injection could be followed over a duration of about 60 s, comparable to most in vivo HP studies. After the HP [^2^H_7_, U-^13^C_6_] Glc bolus, the probability of Glc ^13^C to be transformed into Lac ^13^C is given by the fractional enrichment of its precursors. Using the small pool approximation for the glycolytic intermediates [40], Pyr will quickly follow the Glc fractional enrichment. Consequently, at labeling steady-state, the maximal ^13^C fractional enrichment of Lac would be 70%, if no dilution from other influxes into the brain Lac pool would exist. On the other hand, the probability of ^13^C Lac to leave the brain lactate pool (either by transforming into another molecule in brain tissue or efflux of lactate itself out of the brain) is given by the fractional enrichment of ^13^C Lac that evolves over time. The numerical simulation indicates that at 60 s post bolus of HP [^2^H_7_, U-^13^C_6_] Glc (the time when the acquisition of the HP ^13^C MRS is completed) and for a constant Lac pool size of 4 mmol/kg the fractional enrichment of ^13^C Lac ranges between 6 and 25% (Figure 3c), depending on the CMR_Glc_. Hence, during the 60 s of the HP ^13^C MRS acquisition, the dominating effect of the labeling in the Lac pool is the influx of ^13^C labeling rather than the efflux of the ^13^C. Furthermore, adding the intrinsic physical parameters related to signal losses in the HP MRS experiment (namely, ^13^C relaxation and signal losses due to the RF pulses, see input function in Figure 3f), one can notice that, at the time when the efflux of ^13^C from the Lac pool size becomes more pronounced, the sensitivity to monitor this effect is lost (Figure 3g). Therefore, although the ^13^C Lac signal is clearly measurable due to very strong signal enhancement linked to hyperpolarization, the ^13^C enrichment of the Lac pool remains small over the whole experiment duration, unlike typical non-HP (i.e., thermally polarized) ^13^C MRS experiments, where labeling steady-state with substantial ^13^C fractional enrichment is often reached [41]. Consequently, the probability of a ^13^C labeled Lac to leave the Lac pool is very low at this early time point (compared with the large amount of unlabeled Lac in the same pool). Since the fractional enrichment of Lac is the ratio of ^13^C-labeled Lac over the Lac pool size, for a low ^13^C-labeled Lac concentration, it is expected that the Lac pool size has little impact on the ^13^C efflux from the Lac pool. On the other hand, the ^13^C influx into the Lac pool is equal to the fractional enrichment of the precursor pool (i.e., fractional enrichment of pyruvate) multiplied by the conversion flux of Pyr into Lac, independently of the Lac pool size. These aspects are general to compartmental modeling and can be read in detail in previous reviews [40,41]. The mathematical reflection of this on the labeling turnover curve of Lac is that the initial rising phase of the curve is affected essentially by the influx rate and not the Lac pool size.

To analyze it explicitly in the particular biochemical pathways of brain metabolism of Glc to Lac, we simulated the ^13^C turnover in Lac with typical pool sizes and metabolic fluxes (Figure 3). From Figure 3d, it can be clearly seen that the ^13^C-turnover curve of Lac is hardly sensitive to the Lac pool size during the first 60 s over a 4-fold range of Lac concentrations. In contrast, the same relative change in CMR_Glc_ (i.e., in Lac influx) has a major impact on the first 60 s of the Lac ^13^C turnover curve.

When adding the effects of T_1_ signal relaxation and repetitive RF pulsing on the measured HP signal to the simulation, the relative contributions of CMR_Glc_ and Lac pool size over the first 60 s of the experiment are even more contrasted (Figure 3g,f). In fact, the signal intensity at the later time point at which the Lac pool size is reflected in the Lac ^13^C turnover curve is strongly damped by the HP signal losses. Figure 3g shows the very little impact of the Lac pool size on the measured HP ^13^C Lac signal, while Figure 3f displays the strong sensitivity of the HP ^13^C Lac signal to the CMR_Glc_ value.

The total Lac pool size measured with ^1^H MRS was smaller in mice anesthetized by a combination of medetomidine isoflurane compared to mice anesthetized by the isoflurane solely (~2.4 mM vs. ~4 mM respectively, Appendix A). Consistently, in non-HP [6,6′-^2^H_2_] Glc experiments we could detect labeling of the Lac pool in the ISO group, but not it the MED–ISO group. Interestingly, a different pattern was detected in the HP [^2^H_7_, U-^13^C_6_] Glc experiment, where the labeling of the Lac from the HP ^13^C Glc was 128% higher in the MED–ISO group compared to the ISO group. This discrepancy between the Lac pool size, Lac labeling in [6,6′-^2^H_2_] Glc ^2^H MRS, and Lac labeling in the HP [^2^H_7_, U-^13^C_6_] Glc ^13^C MRS experiment reflects that these two approaches are sensitive to different metabolic parameters. The numerical simulations indicate that the Lac pool size has no impact during the first minute after bolus when the HP [^2^H_7_, U-^13^C_6_] Glc metabolism is measured. As previously suggested [29], the numerical simulations confirm that the first minute of Lac labeling from Glc metabolism is essentially sensitive to changes in CMR_Glc_, which is the dominating metabolic parameter creating the differences in the Lac labeling between the two anesthetic protocols. On the other hand, in the non-HP [6,6′-^2^H_2_] Glc experiment, in which the acquisition times are much longer (about 1 hour), the labeling rate is inversely proportional to the pool size for a given production flux and when labeling steady-state is reached, the measured labeled intensity will be proportional to the total concentration observed in ^1^H MRS [41]. As such, post [6,6′-^2^H_2_] Glc bolus, Lac labeling was observed in the ISO group and not in the MED–ISO group, consistent with the typical higher Lac pool-size in animals anesthetized by isoflurane (Figure 1). Moreover, SNR analysis indicates that in our measurement the noise level was similar in the different experiments (Appendix A). Therefore, if at 64 min, the kinetics of lactate production from [6,6′-^2^H_2_] Glc infusion reached a steady-state, one would expect that the SNR of this lactate signal would correspond to the ratio of the endogenous lactate pool size between ISO and MED–ISO groups (about 2, Appendix A). In our measurements, the SNR of the ^2^H lactate in the ISO group was 8.7 ± 2.6 (Appendix A). Consequently, one would expect that at the steady-state, the corresponding SNR of the ^2^H lactate in the MED–ISO group would be half of this level (i.e., about 4.3). This estimated value is higher than the accepted limit of detection with minimal SNR of 3 [42]. Thus, the absence of the lactate peak is likely not due to lack of sensitivity, but rather because the Lac pool dilution is different. It is possible that, at 64 min post bolus, the ^2^H lactate signal is below the limit of detection.

### 3.2. Brain Consciousness Influence De Novo Lac Production from HP [^2^H_7_, U-^13^C_6_] Glc and the Dynamics of ^2^H-Glx Labelling from Non-HP [6,6′-^2^H_2_] Glc

A recent HP ^13^C MRS study with HP [1-^13^C] Pyr at different brain consciousness states demonstrated that the conversion rate of Pyr to bicarbonate is consistent with the TCA cycle rate determined from non-HP experiments and corresponds to what is expected under different anesthetic conditions [43]. However, when using HP [1-^13^C] Pyr, the conversion rates of Pyr to Lac did not change significantly under different anesthetic conditions [43]. HP Pyr experiments are capable to probe only a specific portion of the Glc metabolism. The Pyr bypasses the Glc transporters and glycolysis by entering the cell via the monocarboxylate transporters (MCTs) and then conversion of ^13^C label from Pyr to Lac. In the HP ^13^C Glc experiment, the production of cerebral [1-^13^C] lactate is a consequence of 12 biochemical steps, including Glc transport, 10 enzymatic steps of glycolysis and Pyr conversion to Lac by lactate dehydrogenase, and should therefore be more sensitive to changes in the Glc metabolic demand.

In the present study, the use of medetomidine combined with isoflurane anesthesia greatly increased the labeling in HP [1-^13^C] Lac compared to the use of isoflurane only. The production of HP [1-^13^C] Lac depends on both the HP [^2^H_7_, U-^13^C_6_] Glc uptake and the metabolic demands (CMR_Glc_). The change for a combination of medetomidine and isoflurane can induce several cerebral and systemic effects that may contribute to the observed differences. Firstly, isoflurane at a clinical concentration (1.4–2.5%) has been associated with a significantly reduced facilitated Glc transport in cells [44,45,46], a reduced local cerebral glucose utilization (LCGU) in all cortical areas [47], and a decrease in Glc metabolic rates due to the inhibition of ATP synthesis in the mitochondria in mice and rats [48,49]. Previous studies using ^18^F-FDG PET also revealed a decrease of the cerebral metabolic rate of this Glc analog under isoflurane [50,51]. Moreover, dose-dependent anesthesia effects of isoflurane were reported for cerebral Glc metabolism in rats, with Glc metabolic rate decreasing up to 41–45% in cortical regions [47,52,53]. As in our experiments, the mouse cortex was well positioned under the sensitive area of the coil, the cortical areas may bring the main contribution to the measured signal, and variations in LCGU between groups may explain some of the differences observed between the two groups. Secondly, isoflurane is known to affect neural activity and neural processing by impacting different factors, such as synaptic transmission, pre-synaptic release, post-synaptic receptors, and membrane potentials [54]. Here, we observed that a larger amount of ^13^C Lac was produced in the case when the functional connectivity was similar to the awake animals (MED–ISO anesthesia) [34]. A higher neuronal functional activity was associated with increased Glc uptake and was suggested to stimulate glycolysis [55,56]. Therefore, it is likely that our results are a reflection of neuronal activity, in line with these studies suggesting that Glc metabolism is linked to neuronal activity, which is decreased with isoflurane anesthesia, and that Lac production can be stimulated when neuronal activity increases [57,58,59].

Interestingly, while the pool size of Glu, Gln, and Glx quantified by the ^1^H spectra was not significantly different between the two anesthetic conditions (Appendix A), the time evolution of the production of ^2^H Glx from [6,6′-^2^H_2_]. Glc suggests that the dynamics of ^2^H Glx labeling is faster in the MED–ISO group, compared to the ISO group (Figure 2e,f). Nonetheless, sum analysis of this metabolite in respect to the HDO signal before the bolus did not report a significant difference after ~1 h post glucose injection (Appendix A). Only a trend of higher Glx labeling in the MED–ISO group compared to ISO group was determined (4.0 ± 1.5 and 3.1 ± 1.0, respectively, *p* = 0.29). One could anticipate that, due to the higher neuronal activity under MED-ISO anesthesia [34], the dynamic of Glx turnover would be faster in the MED–ISO group than in the ISO group. Note that at steady-state, the ^2^H Glx concentrations should be proportional to the endogenous Glx pool-size, which was found similar under both anesthetic conditions (Appendix A). The labeling pattern of Glx following the infusion of [6,6′-^2^H_2_] Glc is a deuterated C4 position after the first TCA cycle turn, which is then lost into water in the following TCA metabolism. Previous thermally polarized ^13^C MRS studies showed that the C4 Glx labeling should reach a steady-state in about 1.5 h [40,41]. Therefore, at about 1 h post bolus, we are close to steady-state, and the trend of Glx signal that was observed in the MED–ISO group is probably related to the faster turnover during the first hour. The detailed analysis of the different Glx turnover pattern under these two anesthetics will require further investigations, and is beyond the scope of this study.

### 3.3. Anesthesia Has a Systemic Effect on Metabolism

Endogenous Lac concentration quantified from ^1^H MRS was found higher in the ISO group, consistent with the known effect of isoflurane, which increases Lac concentration in the brain [60,61,62,63]. Particularly, Horn et al. showed that volatile anesthetics, such as isoflurane, cause a specific increase in extracellular Lac levels in the mouse brain, and that it is not the case with intravenous nor subcutaneous anesthetics, which is coherent with our measurements [61]. The exact mechanism of action of how volatile anesthetic drugs increase endogenous Lac concentrations in the brain remains unknown. Importantly, they demonstrated that Lac formation by isoflurane in the mouse brain was unrelated to neuronal impulse flow [61]. Additionally, the Cr/PCr ratio, related to ATP and ADP balance, was found significantly different between the two groups. Our results suggest a difference in the energetic state between the two anesthetics groups. The shift of phosphocreatine towards creatine in the ISO group may indicate diminished brain energy stores (ATP, PCr), similar to what has been reported on in vivo studies on mice under isoflurane [63], or in post mortem studies of dogs under halothane anesthesia [64]. Additionally, the sum analysis of the metabolites labeled 1 hour post bolus of [6,6′-^2^H_2_] Glc showed labeling in the Lac pool only in the ISO group, as well as a significantly higher HDO labeling in this group. After 1 hour from [6,6′-^2^H_2_] Glc bolus, the metabolites detected by ^2^H MRS are not only a product of cerebral utilization of [6,6′-^2^H_2_] Glc, but also a consequence of the exchange with labeled blood Lac and water resulting from metabolism of Glc in peripheral organs [39]. Thus, the observed significant differences in ^2^H labeling of Lac and water imply a global effect of the anesthesia on the animal’s metabolism. Moreover, blood analysis showed a more than two-fold increase in the total blood lactate concentration in the ISO group (Appendix A). This is in line with previous studies reporting that blood Lac concentration increased by two- to three-fold during inhalation of volatile anesthetics, such as isoflurane or halothane, probably due to the interference of volatile anesthetics with mitochondrial energy metabolism [61,65,66]. The increase of partial pressure of carbon dioxide (pCO_2_), along with the decrease of partial pressure of oxygen (pO_2_) measured in the MED–ISO group, match the results observed in earlier studies, which also reported that isoflurane anesthetics did not induce any change in pO_2_ and pCO_2_ [65]. The present results underline the non-negligible impact of anesthesia, not only on total brain metabolite concentrations, but also on the measured metabolite kinetics. As almost all preclinical MR studies require the use of anesthetics, these effects need to be carefully considered, since they may influence the overall observations. Finally, isoflurane anesthesia is known to affects the dilation of the blood vessels, which influences the cerebral blood flow. In the context of hyperpolarized ^13^C MRS experiment, this would affect mostly the hyperpolarized tracer signal. Analysis of the C1 signal intensity of the HP [^2^H_7_, U-^13^C_6_] Glc indicates that in our setting the tracer signal was not different under the two anesthetic conditions (Appendix A). In the ^2^H MRS measurements, one would expect to observe the influence of the difference in hemodynamics on the [6,6′-^2^H_2_] Glc signal (i.e., the tracer) and the HDO signal as well. In our experiments the [6,6′-^2^H_2_] Glc signal in the sum spectra was not significantly different between the two anesthetic drugs (Appendix A). However, we could measure a trend of higher HDO signal before the bolus in animal anesthetized by isoflurane without reaching statistical significance (Appendix A). Therefore, the analysis of the metabolites ratios for ^2^H MRS data was performed with respect to the mean natural abundance of HDO over all animals. The significantly higher HDO signal post bolus in the ISO group from the summed spectra over the entire duration of the experiment might be influenced by blood flow and indicating non-negligible systemic effect on the observed signals.

### 3.4. Towards Quantitative Analysis of De Novo Lac Production from HP [^2^H_7_, U-^13^C_6_] Glc

An interesting aspect of metabolic imaging modalities is the possibility to measure not only metabolite concentrations, but also metabolic fluxes, i.e., chemical reaction rates in a noninvasive way in vivo. When used as a research tool, metabolic imaging modalities are often developed to obtain quantitative values for the measured metabolic fluxes of interest, i.e., in units of µmol/g/min. However, converting the measured in vivo kinetic signal into quantitative flux values is not a straightforward calibration process and involves typically many steps, from the physics of the signal measurement itself to the biological nature of the tracer conversion into downstream metabolites in the living tissue, which often requires the use of non-linear mathematical models of brain metabolism. Those models need to be detailed enough to properly reflect the biochemical network underlying the tracer conversion, but also simple enough to ensure a robust determination of the derived metabolic fluxes, which typically involves modeling assumptions.

In the particular case of HP MRS studies, the measured intensity of the signal is affected by a series of experimental parameters on top of the metabolic conversion itself, which makes a fully quantitative metabolic interpretation of the results difficult. In the present study for example, the intensity of the measured ^13^C-labeled Lac signal is not only reflecting the ^13^C concentration of Lac in the brain, but is also affected by the T_1_ relaxation and the repetitive RF pulsing on the ^13^C resonance of Lac and its precursors. Although some assumptions can be made on the rate of T_1_ relaxation and the effect of RF pulsing, those additional layers of calibrations and related assumptions make it harder to derive quantitative fluxes, as it was done in the past with thermally polarized MRS experiments or PET tracer experiments. Moreover, in using the surface coil sensitivity profile as signal localization, the spatial dependency of the B_1_ excitation and receive sensitivity are contributions, which cannot be easily isolated without strong assumptions on the signal distribution. Signal localization in HP MRS studies is therefore an important step towards more quantitative metabolic interpretations, but is not a simple process, due to absence of signal recovery and typically large chemical shift range of the metabolites of interest. However, even with a perfectly localized HP signal acquisition protocol, questions remain to be solved, such as the partial saturation of the infused substrate and its recirculation in the blood stream, and the difference in T_1_ relaxation in the blood pool and tissue compartments.

Although a fully quantitative metabolic imaging modality can be an ultimate goal, an understanding of relative contributions of the metabolic parameters of interest to the measured in vivo signal is typically the key point for many research applications. In the clinical context, this is all the more true since reproducible, simple, and robust methods are needed to isolate pathological from normal conditions. Therefore, we made use in the present work of simulation studies based on typical brain metabolic parameters and known experimental parameters in terms of signal relaxation and RF pulsing to analyze the sensitivity of the measured HP ^13^C Lac turnover curve to the metabolic parameters of interest, namely CMR_Glc_ and the Lac pool size. This simulation study shows how the HP ^13^C Lac signal measured in the presented HP [^2^H_7_, U-^13^C_6_] Glc experiment protocol essentially reflects the glycolysis metabolic rate and how it is independent of the Lac pool size itself. Therefore, the HP [^2^H_7_, U-^13^C_6_] Glc experiments provide additional and clearly distinct information to ^1^H MRS and non-HP X-nuclei measurements.

## 4. Materials and Methods

### 4.1. Animal Experimentation

Experiments were performed according to the Swiss law for the protection of animals and were approved by the Veterinary Office of the Canton de Vaud (Service de la consommation et des affaires vétérinaires, VD2353.4d, Vaud, Switzerland). All experiments were conducted according to federal and local ethical laws and complied with the ARRIVE guidelines. Mice were housed with a 12 h light-dark cycle in a temperature- and humidity-controlled animal facility, with free access to food and water. Twelve hours before the experiment, animals were placed in another cage without food.

### 4.2. Study Design

Brain metabolism of HP Glc or non-HP Glc was monitored in 12 hour-fasted male C57BL6/J mice. To monitor the effect of brain consciousness on Glc cerebral metabolism two different anesthetic condition were tested. In the first group, mice were anesthetized by isoflurane solely during the entire duration of the experiments (ISO group). In the second group, mice anesthetic induction was performed with isoflurane for the surgical procedure of placing the catheter for the bolus injection, and then the anesthesia was switched to a combination of medetomidine (Dorbene, Graeub, Switzerland) and low dose isoflurane (MED–ISO group). The duration between the switch in anesthetics and the bolus injection of glucose was chosen to be 60 ± 5 min, to reach a new cerebral metabolic steady-state [67]. In mice that received HP [^2^H_7_, U-^13^C_6_] Glc, single voxel ^1^H MRS measurements were carried out in each mouse 10 min before the bolus and the ^13^C MRS acquisition. An outline of the experimental design is summarized in Figure 4. Overall, 26 mice (age = 17 ± 2 weeks) were employed in this study. Each animal received a single bolus of either HP [^2^H_7_, U-^13^C_6_] Glc or [6,6′-^2^H_2_] Glc. One animal was excluded from this study, as it did not reach any increase in blood glycemia post bolus. Baseline information for the different groups is summarized in Appendix A.

### 4.3. Hyperpolarization

Frozen droplets of a glycerol/water solution 50:50 (*v/v*) containing 2 M [1,2,3,4,5,6,6′-^2^H_7_,U-^13^C_6_]D-Glc (Sigma-Aldrich, Buchs, Switzerland), 25 mM trityl radical OX63 (tris{8carboxyl-2,2,6,6-benzo(1,2-d:5-d)-bis(1,3)dithiole-4-yl-methyl sodium salt) as a polarizing agent (ALBEDA, Copenhagen, Denmark) doped with 1 mM of Gd^3+^ as previously described [68]. Samples were dynamically polarized in a custom-designed 7 T polarizer [69] at 196.68 GHz /1.00 ± 0.05 K for 120 min. Once reaching maximal polarization, it was rapidly dissolved in 5 mL of superheated D_2_O and transferred within 2 s into the separator/infusion pump [70], which was prepositioned inside the magnet bore. The available liquid-state polarization of HP [^2^H_7_, U-^13^C_6_] Glc achieved under these conditions was measured in a separate set of experiments in the separator/infusion pump using a ^13^C RF coil implemented around the pump upon dissolution with D_2_O. Polarization of (29 ± 3%, *n* = 3) was calculated from the ratio between the HP signal of the HP [^2^H_7_, U-^13^C_6_] Glc and its thermal equilibrium signal recorded 15 min after dissolution. For the in vivo measurements, once dissolution completed, a bolus of the solution was automatically infused through a vein catheter (see below) as previously described [71].

### 4.4. Animal Preparation for Magnetic Resonance Scans

Mice were anesthetized using 1.5 ± 0.5% isoflurane (Attane, Minrad, NY, USA) in 60% oxygen using a nose cone. The anesthetized animals were cannulated to place a femoral vein catheter to deliver the Glc bolus. In the ISO group (*n* = 13), mice were kept under 1.3–1.5% isoflurane for the entire duration of the experiment. In the MED–ISO (*n* = 12) group a first subcutaneous (SC) bolus of medetomidine 0.3 mg/kg was administered. Ten minutes after this bolus, isoflurane anesthesia was dropped to 0.25–0.5% and a continuous infusion of medetomidine SC 0.6 mg/kg/h started, as described by Reynaud et al. [72]. A complete blood analysis was performed from the eye of the mouse (retro-orbital bleeding 85 min before Glc bolus). The animals were then transferred to an MRI bed, and their heads were fixed using a stereotaxic system and a bite bar (RAPID Biomedical Inc., Rimpar, Germany) before being entered into the MR scanner. A second blood analysis was performed immediately after the completion of the X-nuclei MRS acquisition (~2 min post bolus of HP [^2^H_7_, U-^1^C_6_] Glc or ~60 min post bolus of non-HP [6,6′-^2^H_2_] Glc) using a PocketChem^TM^ BA analyzer. Animal physiology was monitored during the entire duration of the experiment. Body temperature was monitored by a nonmagnetic rectal probe and maintained at 37.0 ± 0.5 °C by warming the animal with temperature-controlled water circulation (SA instruments Inc. NY, USA). The respiration rate was monitored using a pneumatic pillow sensor (SA Instruments Ins. Stony Brook, NY, USA).

### 4.5. MRS Measurements

All MR measurements were carried out on a Varian INOVA spectrometer (Agilent, Palo Alto, CA, USA) interfaced with a 31-cm horizontal-bore actively shielded 9.4 T magnet (Magnex Scientific, Abingdon, UK).

#### 4.5.1. In Vivo ^1^H Magnetic Resonance Spectroscopy

To evaluate the effect of the two anesthetic conditions on the endogenous concentration of brain metabolites, in vivo ^1^H MRS measurements were performed in the hippocampus in the mice that were injected with HP [^2^H_7_, U-^13^C_6_] Glc bolus. Spectra were acquired using a home-built ^1^H-quadrature/^13^C-single loop surface coil that was placed on top of the mouse head. B_0_ inhomogeneity was corrected using the FASTESTMAP algorithm [73] in a 2 × 2.8 × 2 mm voxel, centered on the hippocampus. ^1^H MRS measurements were acquired using the SPECIAL pulse sequence [74] (TR = 4000 ms, TE = 2.8 ms, in 10 blocks of 16 scans). Absolute metabolite concentrations were quantified using the LCModel [75]. Values with Cramer–Rao lower bounds above 30% were excluded from further analyses.

#### 4.5.2. In Vivo HP ^13^C MRS

Carbon-13 MR measurements were performed using the ^1^H/^13^C surface coil described above that was positioned on top of the mouse head. To improve the detection within the sensitive area of the ^13^C coil B_0_ inhomogeneity was corrected using FASTESTMAP algorithm [73] in a 125 μL voxel. A bolus of 540 μL of 44 ± 10 mM HP [^2^H_7_, U-^13^C_6_] Glc was injected through the vein catheter (245 μL/s) by the automated protocol [71]. Note that the glucose blood levels before injection were within normal physiological concentration (6.5 ± 2 mM). The increase of blood glucose concentration after injection did not exceed the typical values observed in healthy mice after Glc administration (12.5 ± 3 mM) [76]. A series of pulse acquired sequences was then triggered 5.5 s post injection every 1 s for 70 s. To optimize signal-to-noise ratio (SNR) of the expected Glc metabolites in the carboxyl frequency range of the ^13^C spectra, a selective Gaussian pulse (250 μs/40 kHz bandwidth) was centered at 182 ppm resulting in an average nominal 25° flip-angle for the C1 lactate resonance (183.5 ppm) and 1.5° flip-angle for the glucose C1 resonances (92.9 ppm and 96.8 ppm), as previously proposed [28]. At the end of the experiment, 200 μL of the solution remaining in the separator/infusion pump was collected to further compute an estimated blood Glc ^13^C fractional enrichment, with the [^2^H_7_, U-^13^C_6_] Glc concentration of the injected bolus being measured in a high-resolution DRX-400 spectrometer (Bruker Biospin SA/Avancell 800 MHz, TOPSPIN2) using [2-^13^C] acetate 80 mM as reference. For the in vivo acquired spectra, the area under the curve (AUC) of [1-^13^C] Lac (183.5 ppm), [1-^13^C]α-Glc (92.9 ppm), and [1-^13^C]β-Glc (96.8 ppm) were quantified using VNMRJ software by integrating the sum of the ^13^C MRS spectra series, after phase and baseline correction. Overall, 14 mice were employed (ISO group *n* = 8, MED–ISO group *n* = 6).

#### 4.5.3. In Vivo Non-HP ^2^H MRS

A solution of 0.97 ± 0.05 M [6,6′-^2^H_2_] D Glc in a physiological buffer (PBS, Sigma-Aldrich, Buchs, Switzerland) was loaded into the separator/ infusion pump [70] before the animal was entered into the magnet leading to similar bolus dose as previously reported [39]. ^2^H MR measurements were performed using a home-built ^1^H-quadrature/^2^H single loop surface coil that was positioned on top of the mouse head. To improve the detection within the sensitive area of the ^2^H coil, B_0_ inhomogeneity was corrected using the FASTESTMAP [73] algorithm in a 135 μL voxel. A bolus of 350 μL of the [6,6′-^2^H_2_] Glc solution was injected through a vein catheter in a similar automated fashion as for the hyperpolarized infusion (245 μL/s) [71]. To be consistent with the HP ^13^C MRS detection scheme, the ^2^H spectra were acquired using a pulse-acquire sequence, as for the ^13^C experiments. To compensate for B_1_ inhomogeneities, an optimized 90° BIR-4 adiabatic radiofrequency pulse [77] was applied every 2 s in blocks of 32 averages, resulting in 64 s time resolution per measurement point. Approximately 30 min prior to the bolus injection, the endogenous HDO signal was continuously detected (32 blocks of 32 scans). Once the bolus was completed, the ^2^H acquisition was automatically commenced for approximately 60 min (64 blocks of 32 scans). Spectra quantification was performed using AMARES in JMRUI software [78]. To avoid any possible bias due to the slightly larger water content in the FOV of the coil for the animals anesthetized by isoflurane that could result from its vasodilation properties, metabolite ratios were calculated based on the mean water signal before the bolus averaged over all animals. Overall, 11 mice were scanned (ISO group *n* = 5, MED–ISO group *n* = 6).

### 4.6. Numerical Simulations

To analyze the sensitivity of the measured HP ^13^C Lac signal to specific metabolic parameters and support the interpretation of the observed HP time course, simulations were undertaken based on a three-pool model representing brain Glc, Pyr, and Lac ^13^C labeling kinetics (Figure 3). In the Glc metabolic reaction chain in the brain, Pyr is the end product of glycolysis. Pyr is a precursor for oxidative metabolism in the TCA cycle, but is also in conversion with Lac. The model is characterized by three metabolic fluxes: the cerebral metabolic rate of Glc (CMR_Glc_), the conversion between Pyr and Lac through lactate dehydrogenase (V_LDH_), and the exchange between blood Lac and brain tissue Lac (V_out_). For one metabolized molecule of Glc, two molecules of Pyr are produced at the end of the glycolysis. Therefore, for a given CMR_Glc_ flux, the production of Pyr is 2x CMR_Glc_. Applying the metabolic steady-state assumption (constant pool sizes for the metabolites), and from the mass-balance equations of the model, the influx of ^13^C from Pyr into the TCA cycle is also 2x CMR_Glc_ [41]. For the simulations, a typical CMR_Glc_ of 0.5 μmol/g/min was considered. Given the strong activity of lactate dehydrogenase (LDH) allowing fast conversion between Pyr and Lac as compared with the TCA cycle rate [79,80], V_LDH_ was fixed to 50× CMR_Glc_. The exchange with blood Lac V_out_ was fixed to 0.2 μmol/g/min, a value found in a previous non-HP ^13^C MRS study in mice [81]. A realistic Glc labeling input function was used with a 70% ^13^C fractional enrichment reached after a 14 s linear rising period following the bolus tracer injection and blood circulation. A typical Lac pool size of 4 μmol/g was used, while the small Pyr intermediate pool was set to 0.1 μmol/g [28].

The model was expressed in terms of linear differential equations in MATLAB (Version R2018a, The MathWorks, Inc., Natick, MA, USA), and solved using a standard, built-in, ordinary differential equation (ODE) solver.

In a second step, physical parameters influencing the measured HP ^13^C signal were also included in the differential equations of the model. The spin-lattice relaxation rates for Glc R1_gluc_ = 1/14 s^−1^ and for Pyr and Lac R1_lac_ = R1_pyr_ = 1/18 s^−1^ were used [82]. A nominal flip angle of pi/7 was applied every TR = 1 s for Pyr and Lac. For Glc, considering the substantial portion of molecules distributed in the systemic blood circulation away from the coil sensitive volume, an effective TR = 8 s was used to simulate HP Glc signal decay due to repetitive pulsing. The HP Glc signal curve obtained with these parameters represented well the typical measured HP Glc signal.

### 4.7. Statistical Analysis

Statistical analyses were performed using the OriginPro 9.3G software or GraphPad Prism Software (Prism 5.03, GraphPad, La Jolla CA, USA). Unpaired, two-tailed Student’s t-tests were used to compare LGR, blood measurements, age, and neurometabolite concentrations between the two anesthetic conditions. Two-way ANOVA was used to compare the blood glycemia evolution (before and after injection) between the two anesthetic conditions (* *p* < 0.05; ** *p* < 0.01; *** *p* < 0.001). A *p* value of 0.05 was considered significant. All data are presented as mean ± standard deviation, unless otherwise stated.

## 5. Conclusions

HP [^2^H_7_, U-^13^C_6_] Glc reports on de novo Lac synthesis and is sensitive to CMR_Glc_. The level of brain consciousness reached under different anesthesia resulted in variations in cerebral glucose metabolism, but manifested differently in HP and non-HP X-nuclei Glc MRS.

## Figures and Tables

**Figure 1 metabolites-11-00413-f001:**
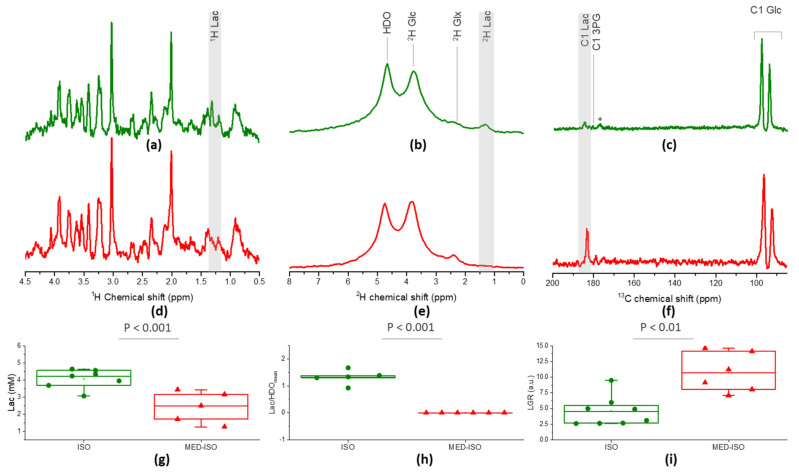
Typical summed spectra measured with ^1^H MRS (**a**,**d**), ^2^H MRS (**b**,**e**) and HP ^13^C MRS (**c**,**f**) under isoflurane solely (‘ISO’, in green) and under medetomidine combined with isoflurane anesthesia (‘MED–ISO’, in red). In (**c**,**f**), the glycolytic intermediate 3-phosphoglycerate (3PG, 179.8 ppm) can be identified in the summed spectra in addition to the C1-Lac at 183.5 ppm. The broad peak at 175 ppm designated by (*) is an impurity in the HP [^2^H_7_, U-^13^C_6_] Glc powder. Significant differences between the two anesthesia groups are found in endogenous Lac concentration measured by ^1^H MRS (**g**), in lactate/HDO_mean_ measured by ^2^H MRS (**h**), and in Lac-to-Glc ratio (LGR) measured by HP ^13^C MRS (**i**).

**Figure 2 metabolites-11-00413-f002:**
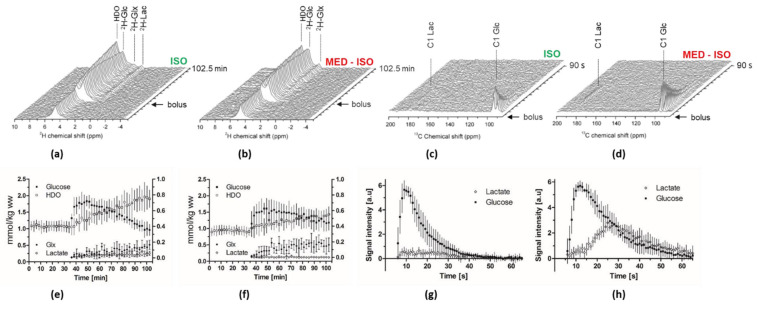
(**a**,**b**) Characteristic dynamic spectra measured in the mouse head under the two different anesthetic protocols. (‘ISO’ or ‘MED–ISO’) following infusion of non-HP [6,6′-^2^H_2_]Glc and the corresponding time courses of water (HDO), Glc (^2^H-Glc), glutamate–glutamine (^2^H-Glx) and lactate (^2^H-Lac) shown for each group (**e**,**f**) respectively, mean ± SD. HDO and Glc refer to the left y-axis while Glx and Lac refer to the right y-axis. (**c**,**d**) Characteristic spectra measured in the mouse head under the two different anesthetic following infusion of HP [^2^H_7_, U-^13^C_6_] Glc and the corresponding time courses of [1-^13^C] Lactate and [1-^13^C]glucose shown for each group (**g**,**h**), mean ± SD; glucose signal normalized to 6, arbitrary units).

**Figure 3 metabolites-11-00413-f003:**
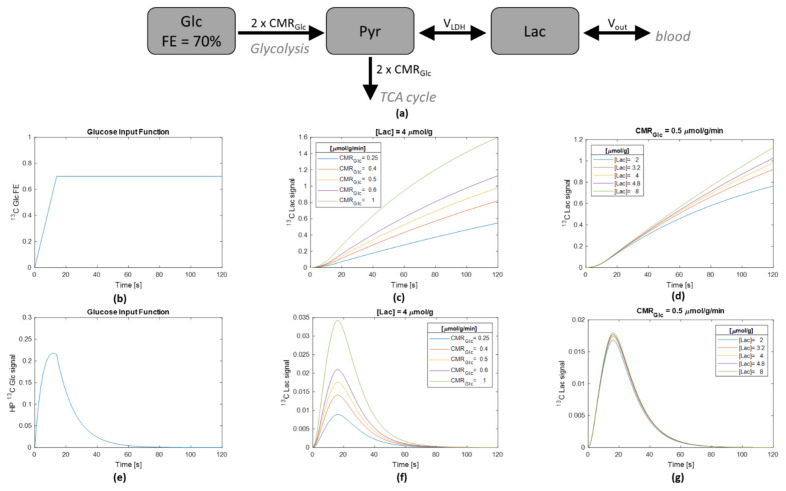
Analysis of the sensitivity of the measured HP ^13^C Lac signal to specific metabolic parameters. Simulations were performed using a three-pool model representing the ^13^C-labeling kinetics of Glc, Pyr, and Lac in the brain (**a**). The model is characterized by three metabolic fluxes: the cerebral metabolic rate of glucose (CMR_Glc_), the conversion between pyruvate and lactate through lactate dehydrogenase (V_LDH_) and the exchange between blood lactate and brain tissue lactate (V_out_). (**b**) A typical step input function for glucose fractional enrichment reaching 70% after 14 s was used. (**c**) The sensitivity of the Lac ^13^C concentration relative to the CMR_Glc_ was assessed by varying the CMR_Glc_ from 0.25 micromole/g/min to 1 µmol/g/min, with a nominal Lac pool size of 4 µmol/g. (**d**) The sensitivity of the Lac ^13^C concentration relative to the Lac pool size was assessed by varying the Lac pool size from 2 µmol/g to 8 µmol/g, with a nominal CMR_Glc_ of 4 µmol/g. In (**e**–**g**), the physical parameters influencing the measured ^13^C HP signal were included in the differential equations of the model (T_1_ decay and repetitive RF pulsing). The ^13^C Lac signal is shown for a varying CMR_Glc_ with a fixed endogenous Lac concentration (**f**) or for varying endogenous Lac concentration with a fixed CMR_Glc_ (**g**).

**Figure 4 metabolites-11-00413-f004:**
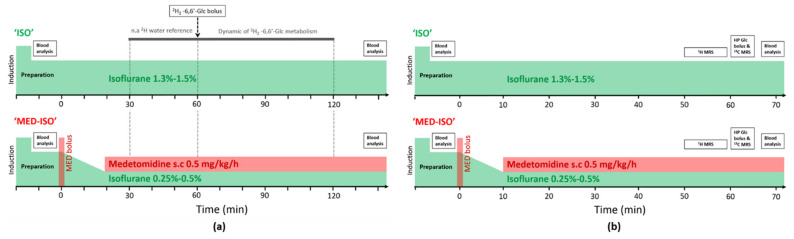
Outline of the experimental design and the anesthetic protocols for (**a**) the experiments using [6,6′-^2^H_2_] Glc and (**b**) the experiments using HP [^2^H_7_, U-^13^C_6_] Glc.

## Data Availability

The data presented in this study are available upon reasonable request from the corresponding author.

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
