# Peer review of "Measuring Glycolytic Activity with Hyperpolarized [2H7, U-13C6] D-Glucose in the Naive Mouse Brain under Different Anesthetic Conditions"

_metabolites, 2021, doi:10.3390/metabo11070413_

Round 1

Reviewer 1 Report

The paper highlights the difference in labeling patterns of produced metabolites due to glucose consumption between X-nuclei MRS of hyperpolarized and non-hyperpolarized glucose.

They show that the discrepancy of acquisition windows between both methods (seconds vs hours) provide insights of different measurements of glycolysis metabolic rate and de novo lactate production.

In addition, authors investigate the impact of 2 widely used anesthetic protocols: isoflurane only (as most of in vivo animal studies) and a combination of isoflurane and medetomidine which preserve functional connectivity and normal glucose consumption.

The paper is very clear and well written. All results and figures support authors claims.

Nonetheless, I have one major comment:

R1: It is well known that isoflurane can strongly affect the cerebral blood flow. I agree that authors used a relatively low dose of isoflurane in ISO condition (between 1.3 and 1.5%). However, they do not comment potential impact of brain perfusion between ISO and MED-ISO groups which could modify the delivery of Glc to brain cells and thus could impact the relative kinetics between both groups. This effect has to be mentioned and discussed in the discussion section.

I have one minor comment:

r1: Page 7, line 4 (line 243 of the full manuscript): “lac labeling” instead of “Lac labeling”.

Reviewer 2 Report

This is commendable effort and of significant impact to interpret the nature of the metabolic detection using two different nuclei using magnetic resonance technology.

My concerns are enumerated below

Major points
    1.    Based on fig 1b and e, looks like glutamate might also be different. A discussion of these results is lacking
    2.    Line 207 - “On such a short duration as compared to the time-scale of the glycolysis rate, not many 13C atoms will be carried from Glc to Pyr and further to Lac.” It would put things in perspective for the readers and the scientific community if the glycolytic rate in these anesthetized mice brains is actually specified and then estimate how much of the infused glucose is likely to end up as lactate. It would also help to use either glycolytic rate or CMR consistently and not interchangeably to improve readability
    3.    Discussion section lines 213-222 need clarification. It seems like the authors are referring to the hyperpolarized lactate signal. However, the way it is written it seems like they are talking in general about the 13C labeled lactate molecules derived from the infused enriched glucose. I think it would be very important to distinguish between the two. As this goes to the crux of the simulation in Results section 2.4
    ⁃    what does “influx of 13C-labeled Lac is directly proportional to the influx into the Lac pool,” mean? Are the authors merely trying to say that the rate of hyperpolarized pyruvate to  lactate conversion is proportional to the rate of thermally labeled conversion of pyruvate to lactate? Does this imply that the system is not in equilibrium during the hyperpolarized experiment ?  
    ⁃    Also statements like “for a low 13C-labeled Lac concentration, it is expected that the Lac pool size has little impact on the 13C efflux from the Lac pool.” . Not clear what 13C efflux from the Lac pool would entitle? - efflux out of the cell / further downstream metabolism ? Also, it seems incorrect as one would expect that the probability of a C13 labeled lactate’s fate would be different based on the total pool size. 
    4.    Line 254 - “Lac labeling was observed in the ISO group and not in the MED-ISO group, consistent with the typical higher Lac pool-size in animals “ - In this particular case, while the rationale of lower lactate FE in the MED+ISO group holds, the lac of detection of 2H lactate is due to the lack of sensitivity. So a discussion of the inherent SNR of the lactate signal under ISO alone and a quick estimation of the SNR required to detect 50% of the signal observed in mice with ISO alone would be pertinent. 
    5.    In general the phrase “exchange of 13C label with the endogenous Lac pool” needs to be avoided, as chemically its not the carbon-13 nuclei that is exchanged, but rather the conversion of 13C labeled pyruvate to lactate. The way this is stated throughout the manuscript needs to be rectified as its confusing. 
    6.    Use of “de novo” lactate production needs better explanation. Is the intent to signify net production of lactate from hyperpolarizied glucose? If so there’s no proof of that offered in this manuscript.  

Minor Points
    1.    Need to define the nomenclature 2H-GLc properly and consistently between 2H and HP 13C studies 
    2.    rephrase sentence spanning 142-144
    3.    replace “labeling in Glc” with “kinetics of infused Glc” 
    4.    Table S2 - clarify glycemic “before” and “after “ to be prior to imaging study and end of imaging study. Denote if its mean +- s.e (or SD?). Also need to clarify fractional enrichment was for 13C and 2H respectively for HP and 2H study
    5.    result section 2.3 - (line 154) talks about de novo synthesis of lactate and towards the end of the result section, line 162, indicates increased labeling of downstream lac - this needs to eb clarified better. 
    6.    Line 200: replace "monitoring of transfer of X-nuclei" with “following the chemical shift difference as the precursor is metabolized” or something like that.
    7.    Line 212: does “where labeling steady-state with substantial 13C fractional enrichment is often reached[40].” Refer to achieving isotopic steady state? If so it should be phrased better
    8.    Line 213: does “13C-labeled Lac to leave the Lac pool “ refer to further downstream metabolism of lactate?
    9.    Line 246: Change “acquisition of the HP 13C Glc metabolism takes place” to “HP 13C GLc metabolism is measured”. Its important to be mindful of what’s observable and what’s biochemically occuring. 
    10.    Line 272 - it would help avoid confusion if “the labeling in [1-13C] Lac” was replaced by “the hyperpolarized Lac signal”
    11.    Please denote in the methods section the duration of infusion of hyperpolarized glucose and the 2H glucose into the mice and define  the notation of 2H and 13C glucose used throughout the text as corresponding to the appropriate isotopomer.

Round 2

Reviewer 1 Report

Thank you for manuscript modifications and the supplemental figure.

Reviewer 2 Report

Thank you for clarifying the concerns that I had raised.  I have no further comments. Great work!

One minor edit for the response corresponding to the sentence 154-156 thus

"Summed analyses (over the 64 min post glucose bolus) of the metabolite ratio showed a significantly higher HDO labeling in the ISO group compared to the MED-ISO group (+28.7 %, p = 0.043) . Moreover, 2H Lac signal was detected only in animals anesthetized by isoflurane alone.